# Dual-Channel Span for Aspect Sentiment Triplet Extraction

**Pan Li[1]**   **Ping Li[1]***   **Kai Zhang[2]**

[1] School of Computer Science, Southwest Petroleum University, Chengdu, China
[2] School of Computer Science and Technology, East China Normal University, Shanghai, China
[1] {lp970717, dping.li}@gmail.com
[2] zk1980@hotmail.com

## Abstract

Aspect Sentiment Triplet Extraction (ASTE) is one of the compound tasks of fine-grained aspect-based sentiment analysis (ABSA), aiming at extracting the triplets of aspect terms, corresponding opinion terms and the associated sentiment orientation. Recent efforts in exploiting span-level semantic interaction have shown superior performance on ASTE task. However, span-based approaches could suffer from excessive noise due to the large number of spans that have to be considered. To ease this burden, we propose a dual-channel span generation method to coherently constrain the search space of span candidates. Specifically, we leverage the syntactic relations among aspect/opinion terms and their part-of-speech characteristics to generate useful span candidates, which empirically reduces span enumeration by nearly a half. Besides, the interaction between syntactic and part-of-speech views brings relevant linguistic information to learned span representations. Extensive experiments on two public datasets demonstrate both the effectiveness of our design and the superiority on ASTE task [1].

## 1 Introduction

Aspect Sentiment Triplet Extraction (ASTE) is a compound task in fine-grained Aspect-Based Sentiment Analysis (ABSA) (Pontiki et al., 2014). It is composed of three fundamental subtasks: *Aspect Term Extraction* (ATE) (Yin et al., 2016; Ma et al., 2019; Chen and Qian, 2020; Li et al., 2020), *Opinion Term Extraction* (OTE) (Yang and Cardie, 2012, 2013; Wan et al., 2020) and *Aspect Sentiment Classification* (ASC) (Wang et al., 2016; Tang et al., 2016; Xue and Li, 2018; Tang et al., 2020; Li et al., 2021). In particular, ASTE aims to extract the sentiment triplet of aspect terms, corresponding opinion terms and their associated sentiment polarity in a given sentence. For example, in the sentence "My vegetable risotto was burnt, and infused totally in a burnt flavor", there are two sentiment triplets, namely, (*"vegetable risotto", "burnt", Negative"*) and (*"flavor", "burnt", Negative"*), where *"vegetable risotto"* and *"flavor"* are aspect terms, *"burnt"* is the opinion term corresponding to the aspect of interest, and *"Negative"* is the sentiment polarity of these two triplets.

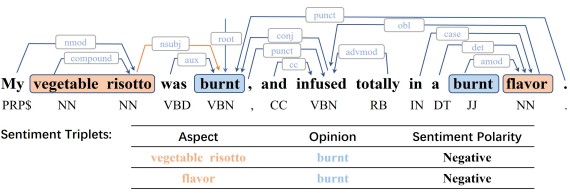

Figure 1: A sentence with dependency tree and part-of-speech in ASTE task.

When the idea of ASTE was first proposed, a two-stage pipeline method (Peng et al., 2020) was developed for this task. However, staged processing scheme often lead to error propagation between subtasks. More than that, opinion terms are generally associated with the aspect target, staged pipeline method breaks this interaction. To address those issue, some end-to-end approaches (Wu et al., 2020a; Xu et al., 2021; Chen et al., 2021b, 2022b) are devised, which attempt to simultaneously extract aspect-opinion pairs and perform sentiment classification by introducing novel tagging schemes. In particular, most of existing end-to-end models (Wu et al., 2020a; Chen et al., 2021b, 2022b) build the interaction between aspect and its corresponding opinion at token-level, i.e., word-to-word interactions. Despite of its efficacy, it is hard to guarantee the consistency of predicted sentiment polarity between multiple word-to-word pairs when many aspects/opinions are expressed using multiple words. On account of this, recent work (Xu et al., 2021; Chen et al., 2022d) adopt span-level interactions in the sentiment triplet structure. Compared with the token-level pairing, span-level in-

---

*Corresponding author: dping.li@gmail.com

teraction is proved to bring significant gains to the model.

However, one prominent problem with span-based methods is that they usually enumerate all spans in a sentence, which will bring about high computational cost and many noises. Specifically, the number of enumerated spans for a sentence of length $n$ is $O(n^2)$, while the number of possible interactions between all opinion and aspect candidate spans is $O(n^4)$ at the later span-pairing stage, implying a lot of invalid aspect/opinion spans and span pairs. Moreover, most of the existing span-based methods model direct interactions between two spans. The high-order interactions are generally overlooked.

To address those issues, we explore the linguistic phenomena in the spans. Our observations are two-fold: **First**, multiple words composed of the span of an aspect/opinion target are generally syntactically dependent, and multiple dependency relations can transmit higher-order interactions between spans. For example, in Figure 1, the aspect term "vegetable risotto" has an intra-span syntactic dependency "compound" and an inter-span dependency "nsubj" with "burnt". On the other hand, the span "flavor" is indirectly related to the first "burnt" (i.e., the one associated with "vegetable risotto") within 2 hops in the syntactic tree. This indirect relation may suggest the relevance of the sentiment polarity of the two span pairs, namely, ("vegetable risotto", "burnt") and ("flavor", "burnt"). In effect, the ground truth of sentiment polarity of these two aspect-opinion pairs are the same, as shown in Figure 1.

**Second**, we also observe that there are some frequent patterns in aspect and opinion spans in terms of part-of-speech. For instance, in many cases aspect terms are noun or noun phrase which we refer to as $NN$ and $(NN - NN)$, respectively. Moreover, it is fairly common that opinion terms are adjective (denoted by $JJ$). As shown in Figure 1, the aspect "vegetable risotto" has the part-of-speech structure $(NN - NN)$, and opinion term "burnt" is $JJ$. Therefore, it is possible to extract the aspect/opinion spans according to the lexical characteristics of the words so as to avoid enumerating all word combination.

Motivated by the two observations, we propose a dual-channel span generation approach for aspect-level sentiment triplet extraction, which we term as Dual-Span. Dual-Span utilizes two relational

graph attention networks (RGAT) to separately learn high-order syntactic dependency between words/spans and linguistic features in constructed part-of-speech relations among words. Then a gating mechanism is adopted to fuse the syntactic and lexical information of span candidates, which helps to enhance the feature representation of spans. On the other hand, instead of enumerating all possible spans, the span candidates are extracted from two channels, i.e., the syntactic dependency relations and part-of-speech based relations, thus largely reducing the noisy information in favor of valid span pairing.

Our main contributions are as follows:

- We devise a dual-channel span generation method for aspect sentiment triplet extraction, which produces a span candidate set much smaller than the greedily enumerated one by leveraging the syntactic dependency and part-of-speech correlation among tokens/spans in a dual-channel manner.

- We construct the intra-span and inter-span relations based on the part-of-speech correlation of spans/words, on top of which the high-order linguistic interactions is able to be captured by relational graph neural networks.

- We combine the syntactic information learned from dependency tree with the part-of-speech information learned from constructed lexical relations to enrich span representation. We conduct extensive experiments on benchmark datasets to evaluate the efficacy and efficiency of the proposed method. The experimental results show that our model Dual-Span outperforms all state-of-the-art methods on the ASTE task.

## 2 Related Work

Aspect-based sentiment analysis (ABSA) (Pontiki et al., 2014; Schouten and Frasincar, 2016; Xue and Li, 2018; Chen et al., 2022a; Trusca and Frasincar, 2023) is fine-grained sentiment analysis. The early work of ABSA was to identify its three sentiment elements (i.e., aspect, opinion, sentiment polarity) as basic tasks: ATE (e.g., (Yin et al., 2016; Ma et al., 2019; Chen and Qian, 2020; Li et al., 2020), OTE (Yang and Cardie, 2012, 2013; Wan et al., 2020)) and ASC (e.g., (Wang et al., 2016; Tang et al., 2016; Xue and Li, 2018; Du et al., 2019;

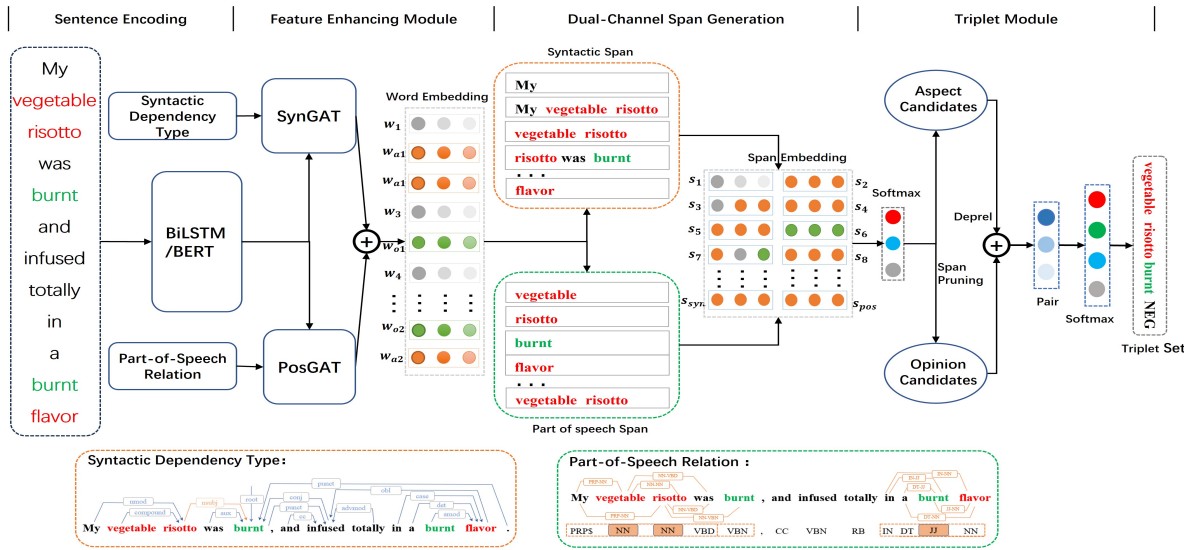

Figure 2: The overall architecture of our Dual-Span.

Li et al., 2021; Brauwers and Frasincar, 2023)). Subsequently, some studies began to consider multiple sentiment element composite tasks in order to better understand fine-grained sentiment analysis: *aspect term polarity co-extraction* (APCE) (Li and Lu, 2017; He et al., 2019; Li et al., 2019), *Aspect-Opinion Pair Extraction* (AOPE) (Zhao et al., 2020; Wu et al., 2020a; Gao et al., 2021; Chakraborty et al., 2022) and *Aspect Category Sentiment Analysis* (ACSA) (Schmitt et al., 2018; Hu et al., 2019; Cai et al., 2020; Liu et al., 2021).

Some recent works started to consider the integrity among the three sentiment elements and thus proposed the ASTE task. A diversity of techniques were proposed for it: two-stage pipeline (Peng et al., 2020), multi-task unified framework (Li et al., 2019; Zhang et al., 2020; Yan et al., 2021), multi-round machine reading comprehension method (Mao et al., 2021; Chen et al., 2021a; Liu et al., 2022) and end-to-end method (Wu et al., 2020a; Xu et al., 2020; Chen et al., 2021b, 2022c; Xu et al., 2021; Chen et al., 2022d). The span-level based approaches adopt end-to-end implementation. For instance, Span-ASTE (Xu et al., 2021) enumerates aspect and viewpoint spans and directly exploits their interaction to solve ASTE tasks, while SBN (Chen et al., 2022d) proposed a span-level bidirectional network that enumerates all possible spans as input, and completes the ASTE task by designing two decoders and adopting inference strategies. Despite that, it still remains an open challenge to improve the search efficiency and feature representation for

the span of sentiment triplets.

## 3 Proposed Framework

In this section, the overall architecture of our proposed model Dual-Span is shown in Figure 2, which consists of four main components: sentence encoding, feature enhancing module, dual-channel span generation and triplet module.

### 3.1 Task Definition

For a sentence $X = \{w_1, w_2, \ldots, w_n\}$ of length $n$, the ASTE task is to extract the set of aspect sentiment triplets $\mathcal{T} = \{(a, o, s)_m\}_{m=1}^{|\mathcal{T}|}$ from the given sentence $X$, where $a$, $o$ and $s \in \{POS, NEU, NEG\}$ represent the aspect term, opinion term and sentiment polarity, respectively. $|\mathcal{T}|$ is the number of sentiment triplets contained sentence $X$.

### 3.2 Sentence Encoding

To obtain contextual representations for each word, we explore two sentence encoding methods, namely, BiLSTM and BERT.

**BiLSTM** We first use the GloVe (Pennington et al., 2014) embedding to get the embedding matrix $E \in \mathbb{R}^{|V|*d_w}$ of the corpus, where $|V|$ represents the vocabulary size, and $d_s$ represents the embedding dimension. For the embedding tokens $E_x = \{e_1, e_2, \ldots, e_n\}$ in the sentence, we use BiLSTM to get its hidden representation $H = \{h_1, h_2, \ldots, h_n\}$, where $h \in \mathbb{R}^{2d_n}$ is obtained by splicing the hidden state $\overrightarrow{h} \in R^{d_n}$ generated by

forward LSTM and the hidden state $\overleftarrow{h} \in R^{d_n}$ generated by backward LSTM:

$$h = [\overrightarrow{h}; \overleftarrow{h}] \quad (1)$$

**BERT** An alternative approach is to utilize BERT (Devlin et al., 2019) as the sentence encoder to generate contextualized word representations. Given a sentence $X = \{w_1, w_2, \ldots, w_n\}$ with $n$ words, the hidden representation sequence $H = \{h_1, h_2, \ldots, h_n\}$ is the output of the encoding layer of BERT at the last transformer block.

### 3.3 Feature Enhancing Module

As aforementioned, spans (or intra-span words) involve syntactical dependency and part-of-speech correlation, therefore incorporating those information into feature representations can be beneficial for span pairing and sentiment prediction. To capture the high order dependency relations, here we devise a graph neural network based method to encode the syntactic dependency and part-of-speech relations of intra- and inter-spans in high orders. In particular, we construct the part-of-speech relational graph (corresponding to a multi-relation matrix as shown in Figure 3 (b)). Then we apply two relational graph attention networks to learn the high order interactions between words on syntactic dependency tree of the sentence in question and constructed part-of-speech graph, respectively.

#### 3.3.1 Part-of-speech Graph Construction

The goal of part-of-speech graph construction is to characterize the word formation patterns of aspect and opinion terms so as to better identify the possible spans. Specifically, we adopt the following three rules to construct the part-of-speech graph $G^{Pos} = (V, R^{Pos})$ of a given sentence $X$. First, following previous work (Chakraborty et al., 2022), assuming that aspect terms are usually nouns and opinion terms are usually adjectives, we can define part-of-speech relations based on part-of-speech tags $NN$ or $JJ$. In particular, we consider the relations between words in a given window that contains words tagged with $NN$ or $JJ$. Therefore, a relational edge $R^{Pos}_{i,j}$ of $G^{Pos}$ is defined for two words $i$ and $j$ as the combination of part-of-speech tags of the two words, whose representation vector is $r^p_{i,j} \in \mathbb{R}^{d_p}$, where $d_p$ is the dimension of part-of-speech combination embedding. Besides, we consider the special syntactic relation $nsubj$, since opinion terms are usually directly used to

modify aspect terms, leading to better extraction of aspect-opinion pairs. Finally, for each word's part-of-speech, we add a self-loop relational edge to itself, as the diagonal elements shown in Figure 3.

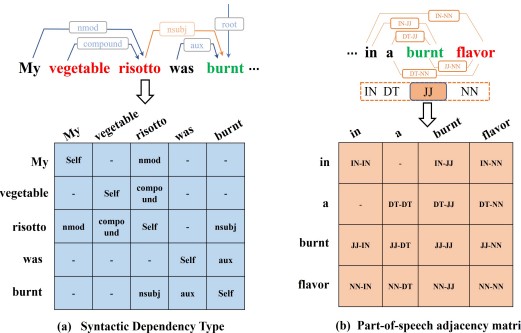

(a) Syntactic Dependency Type    (b) Part-of-speech adjacency matrix

Figure 3: An example sentence with dependency tree and part-of-speech adjacency matrices in ASTE task.

On the other hand, the syntactic dependency graph $G^{Syn} = (V, R^{Syn})$ is constructed according to the dependency parsing tree, where edges are represented by syntactic relation types. Moreover, we define the self-dependency for each word. So for a given sentence of length $n$, the syntactic relation between words $w_i$ and $w_j$ is denoted as $R^{Syn}_{i,j}$, whose corresponding vectorization representation is denoted as the vector $r^s_{i,j} \in \mathbb{R}^{d_s}$, where $d_s$ is the dimension of syntactic relation embeddings.

#### 3.3.2 High-order Feature Learning with Relational Graph Attention Network

Next, we use relational graph attention networks (RGAT) to capture the multiple types of linguistic features and high-order interaction between spans/words on syntactic dependency graph and part-of-speech graph, respectively. Moreover, we use two graph attentional network based modules, namely, SynGAT and PosGAT to learn syntactic dependency graphs and part-of-speech graphs, respectively, which will distinguish between various syntactic relationships and part-of-speech relationships when calculating the attention weight between nodes. In particular, following previous work (Bai et al., 2021), we denote two specific relations on each edge by $r^s_{i,j}$ and $r^p_{i,j}$, respectively. Specifically, for the $i-th$ node, the update process is as follows:

$$h_i^{syn}(l) = \|_{z=1}^Z \sigma\left(\sum_{j \in \mathcal{N}(i)} \hat{\alpha}_{i,j}^{lz}\left(W_{s1}^{lz}h_j^{syn}(l-1) + W_{s2}^l r_{i,j}^{syn}\right)\right) \quad (2)$$

$$h_i^{pos}(l) = \|_{z=1}^Z \sigma\left(\sum_{j \in \mathcal{N}(i)} \hat{\beta}_{i,j}^{lz}\left(W_{p1}^{lz}h_j^{pos}(l-1) + W_{p2}^l r_{i,j}^{pos}\right)\right) \quad (3)$$

where $W_{s2}^l \in \mathbb{R}^{\frac{d}{z} \times d}$ and $W_{p2}^l \in \mathbb{R}^{\frac{d}{z} \times d}$ are parameter matrices. $z$ denotes the number of attention heads, and $\sigma$ is the sigmoid activation. $\mathcal{N}_i$ is the set of immediate neighbors of node $i$. $\hat{\alpha}_{i,j}^{(lz)}$, $\hat{\beta}_{i,j}^{(lz)}$ are the normalized attention coefficients for the $z$-th head at the $l$-th layer.

To fuse syntactic dependency and part-of-speech relation features, we introduce a gating mechanism (Cho et al., 2014) to merge the two views as follows:

$$g = \sigma \left( W_g \left[ h^{syn} : h^{pos} \right] + b_g \right) \quad (4)$$

$$h = g \circ h^{syn} + (1 - g) \circ h^{pos} \quad (5)$$

where $\circ$ is element-wise product operation. $[h^{syn} : h^{pos}]$ is the concatenation of $h^{syn}$ and $h^{pos}$, and $W_g$ and $b_g$ are model parameters. This way, $g$ is learned to optimize the feature fusion.

### 3.4 Dual-Channel Span Generation

In this section, we propose a dual-channel span generation module, which consists of two parts: dual-channel span generation and span classification.

#### 3.4.1 Dual-Channel Span Generation

**Syntactic Span Generation** Given a sentence $X$ whose syntactic dependency graph is $G^{Syn} = (V, R^{Syn})$, if there is a dependency edge $e_{ij}$ between words $w_i$ and $w_j$, then all words positioned between them are considered to be a span $\mathbf{s}_{i,j}^{syn}$. In particular, self-dependent edges represents spans of length $L_s = 1$. We define the representation of $\mathbf{s}_{i,j}^{syn}$ as follows

$$\mathbf{s}_{i,j}^{syn} = [h_i : h_j : f_{width}(i,j)], \text{ if } e_{i,j} = 1 \quad (6)$$

where $f_{width}(i,j)$ denotes trainable embedding of span length (i.e., $j - i + 1$). $e_{i,j} = 1$ suggests that there is an edge between $w_i$ and $w_j$.

**Part-of-speech Span Generation** For a given sentence $X = \{w_1, w_2, \ldots, w_n\}$, if the part-of-speech tag of word $w_o$ is $NN$ or $JJ$, the words in a predefined window will be exhaustively enumerated and then the enumeration is further combined with central word $w_o$ to form spans. The part-of-speech induced span $\mathbf{s}_{k,l}^{pos}$ can be represented as:

$$\mathbf{s}_{k,l}^{pos} = [h_k : h_l : f_{width}(k,l)],$$
$$\text{if } pos_o = NN \text{ or } JJ, \text{ and } o \in [k, l] \quad (7)$$

where $f_{width}(k,l)$ refers to the trainable embedding of span length.

Finally, we merge the two types of span candidates: $S = \mathbf{s}_{i,j}^{syn} \cup \mathbf{s}_{k,l}^{pos}$.

Compared to exhaustive enumeration on the whole sentence in previous span-based approaches, whose time complexity of enumerated spans is $O(n^2)$, for a sentence of length n. However, in our syntactic span generation, the parsing tree containing $2n$ edge dependencies (Qi et al., 2020) (including self-dependent edges), so the number of generated spans is $O(2n)$. On the other hand, the statistics shows that in the benchmark datasets, there are about 2.5 part-of-speech $NN$ and $JJ$ in each sentence on average. Therefore, in the part-of-speech span generation procedure, the number of span candidates is $O(2.5 S_{window}(S_{window} - 1)) \leq n$, where $S_{window}$ is the window size to restrict span length and generally set to be a small value (e.g., $S_{window} = 3$ in our experiments). That is, the time complexity of our method to generate the span is $O(n)$, which significantly reduce the span candidate size.

#### 3.4.2 Span Classification

After obtaining the span candidates $S$, we further narrow down the pool of possible spans by leveraging two auxiliary tasks, namely, ATE and OTE tasks. Specifically, all span candidates in $S$ will be classified into one of the three categories:{Aspect, Opinion, Invalid} by a span classifier. Next, $nz$ spans are singled out with higher prediction scores $\Phi_{aspect}$ or $\Phi_{opinion}$, where $z$ is the threshold hyper-parameter and $\Phi_{aspect}$ and $\Phi_{opinion}$ are obtained by

$$\Phi_{aspect}(\mathbf{s}_{i,j}) = \text{softmax}\left(\text{FFNN}_{t=aspect}(\mathbf{s}_{i,j})\right) \quad (8)$$

$$\Phi_{opinion}(\mathbf{s}_{i,j}) = \text{softmax}\left(\text{FFNN}_{t=opinion}(\mathbf{s}_{i,j})\right) \quad (9)$$

where FFNN denotes a feed-forward neural network with non-linear activation.

### 3.5 Triplet Module

Based on the shrinked candidate pool of aspect and opinion terms, the aspect candidate $\mathbf{s}_{a,b}^a \in S^a$ and opinion candidate $\mathbf{s}_{c,d}^o \in S^o$ are paired and represented as

$$\mathbf{g}_{\mathbf{s}_{a,b}^a, \mathbf{s}_{c,d}^o} = [\mathbf{s}_{a,b}^a : \mathbf{s}_{c,d}^o : r_{ab,cd}^s : f_{distance}(a,b,c,d)]. \quad (10)$$

where $f_{distance}(a, b, c, d)$ denotes trainable embeddings of span length. $r_{ab,cd}^s$ is a trainable embedding vector which is the average pooling of the dependency vectors between words $ab$

and $cd$. Additionally, since opinions are more likely to modify the aspects that match them, we consider the dependency relationship $r^s_{ab,cd} \in R^{Syn}$ between them. Then, sentiment classification is performed for the obtained span pairs, where the sentiment types are defined as $r \in R = \{$Positive, Negative, Neutral, Invalid$\}$ Formally, the triplet prediction is written as

$$P\left(r \mid \mathbf{s}^a_{a,b}, \mathbf{s}^o_{c,d}\right) = \mathrm{softmax}\left(\mathrm{FFNN}_r\left(\mathbf{g}_{\mathbf{s}^a_{a,b},\mathbf{s}^o_{c,d}}\right)\right) \quad (11)$$

### 3.6 Training objective

The loss function for training is defined as the sum of the negative log-likelihoods from the span-pair classification in the span-classification and triplets modules:

$$\begin{aligned}
\mathcal{L} = &- \sum_{\mathbf{s}_{i,j} \in S} \log P\left(\hat{t}_{i,j} \mid \mathbf{s}_{i,j}\right) \\
&- \sum_{\mathbf{s}^t_{a,b} \in S^a, \mathbf{s}^o_{c,d} \in S^o} \log P\left(\hat{r} \mid \mathbf{s}^a_{a,b}, \mathbf{s}^o_{c,d}\right)
\end{aligned} \quad (12)$$

where $\hat{t}_{i,j}$ and $\hat{r}_{i,j}$ are the ground truth labels for span $\mathbf{s}_{i,j}$ and span-pair $(\mathbf{s}^a_{a,b}, \mathbf{s}^o_{c,d})$, respectively. $S$, $S^a$ and $S^o$ are the final span representation, pruned aspect and opinion candidate pools in dual-channel span generation, respectively.

## 4 Experiments

### 4.1 Datasets

To verify the effectiveness of our proposed model, we conduct experiments on four public datasets, i.e., Lap14, Res14, Res 15 and Res16, which come from the sentiment evaluation benchmarks SemEval 2014 (Pontiki et al., 2014), SemEval 2015 (Pontiki et al., 2015) and SemEval 2016 (Pontiki et al., 2016), respectively. Moreover, the four datasets have two versions: ASTE-Data-v1(D1 for short) (Peng et al., 2020) and ASTE-Data-v2(D2 for short) (Xu et al., 2020). Statistics for public datasets are shown in the appendix A.1.

### 4.2 Experimental Setting

We initialize word embedding with two different encoders: BiLSTM-based and BERT-based encoders. The hidden dimension of BiLSTM-based encoder is set to 300 with dropout rate 0.5. To alleviate overfitting, the input embedding dropout rate is 0.7. For the proposed model, we use the AdamW optimizer (Loshchilov and Hutter, 2017) with a

learning rate of 1e-3 in the training. In the implementation of BERT-based encoding, the model parameters are optimized using Adamw with a maximum learning rate of 5e-5 and weight decay of 1e-2. We run the model for 20 training epochs. For other parameter groups, the same parameter settings are used for both embedding initialization schemes. The maximum span length $L_s$ is fixed to 8, the span pruning threshold $z$ is set to 0.5, and the part-of-speech window $S_{window}$ is 3. We choose the best model parameters based on the F1 score on the validation set and report the average of the results for 5 different random seeds.

### 4.3 Baselines

We compare our model to the following state-of-the-art methods:

- **Pipeline:** including CMLA+ (Wang et al., 2017), RINANTE+ (Dai and Song, 2019), Li-unified-R (Li et al., 2019), Peng-two-stage (Peng et al., 2020) and IMN+IOG (Wu et al., 2020b).

- **End-to-end:** OTE-MTL (Zhang et al., 2020), JET (Xu et al., 2020), GTS-CNN, GTS-BiLSTM, GTS-BERT (Wu et al., 2020a), S$^3$E$^2$ (Chen et al., 2021b), BART-ABSA (Yan et al., 2021), MTDTN (Zhao et al., 2022), EMC-GCN (Chen et al., 2022c). These approaches are end-to-end models that include a unified grid tagging scheme and a position-aware tagging scheme.

- **MRC:** Dual-MRC (Mao et al., 2021), BMRC (Chen et al., 2021a), COM-MRC (Zhai et al., 2022). All these method are based on the framework of machine reading comprehension.

- **Span-based:** Span-ASTE (Xu et al., 2021), SBN (Chen et al., 2022d). Span-based models consider all possible spans in a sentence and match aspect terms with opinion terms in an end-to-end manner.

### 4.4 Main Results

We conduct experiments on the two versions of four benchmark datasets, i.e., D1 and D2, whose results are shown in Table 1 and Table 2, respectively. As can be seen from the two tables, under the comprehensive performance indicator F1, the proposed Dual-span consistently outperforms

| Category | Model | Lap14 | | | Res14 | | | Res15 | | | Res16 | | |
|---|---|---|---|---|---|---|---|---|---|---|---|---|---|
| | | P | R | F1 | P | R | F1 | P | R | F1 | P | R | F1 |
| **BiLSTM** Pipeline | Li-unified-R(2019) | 42.25 | 42.78 | 42.47 | 41.44 | 68.79 | 51.68 | 43.34 | 50.73 | 46.69 | 38.19 | 53.47 | 44.51 |
| | Peng-two-stage(2019) | 48.62 | 45.52 | 47.02 | 58.89 | 60.41 | 59.64 | 51.07 | 46.04 | 48.71 | 59.25 | 58.09 | 58.67 |
| | IMN+IOG(2020) | 49.21 | 46.23 | 47.68 | 59.57 | 63.88 | 61.65 | 55.24 | 52.33 | 53.75 | - | - | - |
| End-to-end | GTS-CNN(2020) | 55.93 | 47.52 | 51.38 | 70.79 | 61.71 | 65.94 | 60.09 | 53.57 | 56.64 | 62.63 | 66.98 | 64.73 |
| | GTS-BiLSTM(2020) | 59.42 | 45.13 | 51.30 | 67.28 | 61.91 | 64.49 | 63.26 | 50.71 | 56.29 | 66.07 | 65.05 | 65.56 |
| | S³E²(2021) | 59.43 | 46.23 | 52.01 | 69.08 | 64.55 | 66.74 | 61.06 | 56.44 | 58.66 | 71.08 | 63.13 | 66.87 |
| Span-based | Span-ASTE(2021) | 59.85 | 45.67 | 51.80 | 72.52 | 62.43 | 67.08 | 64.29 | 52.12 | 57.56 | 67.25 | 61.75 | 64.37 |
| Ours | Dual-Span | 60.14 | 47.65 | **53.53** | 74.82 | 61.97 | **68.19** | 64.71 | 57.37 | **60.76** | 73.47 | 62.46 | **67.49** |
| **BERT** End-to-end | MTDTN(2022) | 61.98 | 54.71 | 58.12 | 70.00 | 71.78 | 70.88 | 59.03 | 62.68 | 60.80 | 69.04 | 69.98 | 69.51 |
| | EMC-GCN(2022) | 61.46 | 55.56 | 58.32 | 71.85 | 72.12 | 71.98 | 59.89 | 61.05 | 60.38 | 65.08 | 71.66 | 68.18 |
| MRC-based | BMRC(2021) | - | - | 57.83 | - | - | 70.01 | - | - | 58.74 | - | - | 67.49 |
| | COM-MRC(2022) | 64.73 | 56.09 | 60.09 | 76.45 | 69.67 | 72.89 | 68.50 | 59.74 | 63.65 | 72.80 | 70.85 | 71.79 |
| Ours | Dual-Span | 64.50 | 58.59 | **61.36** | 77.55 | 73.52 | **75.47** | 67.66 | 66.14 | **66.85** | 72.44 | 73.47 | **72.94** |

Table 1: Experimental results on dataset D1, including two versions of BiLSTM and BERT. All baseline results are from the original paper. Best results are in bold and the second best are underlined.

| Category | Model | Lap14 | | | Res14 | | | Res15 | | | Res16 | | |
|---|---|---|---|---|---|---|---|---|---|---|---|---|---|
| | | P | R | F1 | P | R | F1 | P | R | F1 | P | R | F1 |
| Pipeline | CMLA+(2017) | 30.09 | 36.92 | 33.16 | 39.18 | 47.13 | 42.79 | 34.56 | 39.84 | 37.01 | 41.34 | 42.10 | 41.72 |
| | RINANTE+(2019) | 21.71 | 18.66 | 20.07 | 31.42 | 39.38 | 34.95 | 29.88 | 30.06 | 29.97 | 25.68 | 22.30 | 23.87 |
| | Li-unified-R(2019) | 40.56 | 44.28 | 42.34 | 41.04 | 67.35 | 51.00 | 44.72 | 51.39 | 47.82 | 37.33 | 54.51 | 44.31 |
| | Peng-two-stage(2019) | 37.38 | 50.38 | 42.87 | 43.24 | 63.66 | 51.46 | 48.07 | 57.51 | 52.32 | 46.96 | 64.24 | 54.21 |
| End-to-end | OTE-MTL (2020) | 49.53 | 39.22 | 43.42 | 62.00 | 55.97 | 58.71 | 56.37 | 40.94 | 47.13 | 62.88 | 52.10 | 56.96 |
| | JET-BERT°$_{M=6}$(2020) | 55.39 | 47.33 | 51.04 | 70.56 | 55.94 | 62.40 | 64.45 | 51.96 | 57.53 | 70.42 | 58.37 | 63.83 |
| | GTS-BERT(2020) | 57.52 | 51.92 | 54.58 | 70.92 | 69.49 | 70.20 | 59.29 | 58.07 | 58.67 | 68.58 | 66.86 | 67.58 |
| | BART-ABSA(2021) | 61.41 | 56.19 | 58.69 | 65.52 | 64.99 | 65.25 | 59.14 | 59.38 | 59.26 | 66.60 | 68.68 | 67.62 |
| | EMC-GCN(2022) | 61.46 | 55.56 | 58.32 | 71.85 | 72.12 | 71.98 | 59.89 | 61.05 | 60.38 | 65.08 | 71.66 | 68.18 |
| MRC-based | Dual-MRC(2021) | 57.39 | 53.88 | 55.58 | 71.55 | 69.14 | 70.32 | 63.78 | 51.87 | 57.21 | 68.60 | 66.24 | 67.40 |
| | BMRC(2021) | 70.55 | 48.98 | 57.82 | 75.61 | 61.77 | 67.99 | 68.51 | 53.40 | 60.02 | 71.20 | 61.08 | 65.75 |
| | COM-MRC(2022) | 62.35 | 58.16 | 60.17 | 75.46 | 68.91 | 72.01 | 68.35 | 61.24 | 64.53 | 71.55 | 71.59 | 71.57 |
| Span-based | Span-ASTE-BERT(2021) | 63.44 | 55.84 | 59.38 | 72.89 | 70.89 | 71.85 | 62.18 | 64.45 | 63.27 | 69.40 | 71.17 | 70.26 |
| | SBN(2022) | 65.68 | 59.88 | 62.65 | 76.36 | 72.43 | 74.34 | 69.93 | 60.41 | 64.82 | 71.59 | 72.57 | 72.08 |
| Ours(BERT) | Dual-Span | 67.14 | 62.13 | **64.49** | 77.01 | 74.00 | **75.47** | 67.97 | 66.34 | **67.13** | 73.56 | 73.48 | **73.49** |

Table 2: Experimental results on dataset D2, all baselines are from the original text. Best results are in bold and the second best are underlined.

all baselines both for BiLTSM encoder and BERT encoder. Moreover, our model achieves the superior performance in precision and/or recall in most cases. On the other side, the experimental results suggest that non-pipeline methods (i.e., End-to-end, MRC-based, Span-based) are better than pipeline methods, which should be attributed to the fact that the pipeline methods do not consider the correlation between sentiment elements, thus leading to error propagation between stages. It is noteworthy that among tagging based end-to-end methods, some methods that employ syntactic structure of the sentence such as S3E2, MTDTN and EMC-GCN generally outperform the methods that only learn tagging information (e.g., OTE-MTL, GTS and JET), suggesting that the syntactic features of sentences are meaningful for triplet representation. In particular, our end-to-end Dual-Span model outperforms all end-to-end based methods and span-based methods Span-ASTE, SBN, which can be attributed to the fact that our method not only uti-

| Model | Lap14 | Res14 | Res15 | Res16 |
|---|---|---|---|---|
| w/o Dual-RGAT | 59.95 | 69.77 | 63.4 | 70.12 |
| w/o SynGAT | 61.62 | 70.12 | 63.02 | 70.13 |
| w/o PosGAT | 64.23 | 71.32 | 62.42 | 70.89 |
| Transformer | 63.16 | 72.14 | 63.32 | 72.49 |
| Dual-GAT | 61.71 | 72.67 | 64.23 | 72.13 |
| w/o SynSpan | 62.77 | 73.27 | 63.86 | 70.89 |
| w/o PosSpan | 64.39 | 73.46 | 65.51 | 71.34 |
| **Dual-Span** | **64.49** | **75.47** | **67.13** | **73.49** |

Table 3: Experimental results of ablation study.

lizes the syntactic relationship and other linguistic features of sentences for span representation learning, but reduce the noise for span generation and pairing, which can facilitate valid span pairing. Specifically, the F1 score of Dua-Span on datasets D1 and D2 outperforms over other state-of-the-art models by about 2% on average.

| D2 | Model | ATE | | | OTE | | |
|---|---|---|---|---|---|---|---|
| | | P | R | F1 | P | R | F1 |
| | GTS-BERT | 76.63 | 82.68 | 79.53 | 76.11 | 78.44 | 77.25 |
| Lap14 | Span-ASTE | 81.48 | 86.39 | 83.86 | 83.00 | 82.28 | 82.63 |
| | **Dual-Span** | 80.67 | 87.92 | **84.14** | 78.96 | 84.07 | 81.44 |
| | GTS-BERT | 78.12 | 85.64 | 81.69 | 81.12 | 88.24 | 84.53 |
| Res14 | Span-ASTE | 83.56 | 87.59 | 85.50 | 82.93 | 89.67 | 86.16 |
| | **Dual-Span** | 83.19 | 89.95 | **86.44** | 83.38 | 88.99 | 86.10 |
| | GTS-BERT | 75.13 | 81.57 | 78.21 | 74.96 | 82.52 | 78.49 |
| Res15 | Span-ASTE | 78.97 | 84.68 | 81.72 | 77.36 | 84.86 | 80.93 |
| | **Dual-Span** | 81.49 | 84.16 | **82.80** | 78.17 | 87.25 | **82.46** |
| | GTS-BERT | 75.06 | 89.42 | 81.61 | 78.99 | 88.71 | 83.57 |
| Res16 | Span-ASTE | 79.78 | 88.50 | 83.89 | 82.59 | 90.91 | 86.54 |
| | **Dual-Span** | 78.45 | 90.24 | **83.93** | 82.24 | 88.26 | 85.14 |

Table 4: Experimental results of ATE and OTE tasks on dataset D2.

## 4.5 Model Analysis

### 4.5.1 Ablation Study

To further explore the effectiveness of different modules in Dual-Span, we conduct ablation experiments on the D2 dataset. Table 3 shows the experimental results in terms of F1 scores in D2. W/o SynGAT and w/o PosGAT denote the removal of syntactic graph attention network (SynGAT) and part-of-speech graph attention network (PosGAT), respectively, while W/o Dual-RGAT denotes the removal of both SynGAT and PosGAT. We also compare our approach with unitary graph attention networks Dual-GAT that performs attention convolution on syntactic dependency graphs and part-of-speech graphs respectively without distinguishing edge types. By comparing w/o SynGAT, w/o PosGAT, w/o Dual-RGAT and Dual-Span, we observe that both the dependency relationship and part-of-speech features of the sentence are informative to the representation of spans. In particular, the syntactic structural feature and part-of-speech information can be complementary. This is manifested by the outperformance of Dual-RGAT over Transformer and Dual-GAT. When removing the syntactic span generation module (corresponding to w/o SynSpan) or part-of-speech span generation module (corresponding to w/o PosSpan), the performance is also degraded. This observation illustrates that span candidate size can be effectively reduced. Overall, each module of our Dual-span contributes to the overall performance of the ASTE task.

### 4.5.2 Effectiveness of Dual-Span in Span Generation

We use two subtasks, namely, ATE and OTE of ABSA, to explore the effectiveness of dual-channel span Generation strategy. We evaluate our model

| Model | Lap14 | Res14 | Res15 | Res16 |
|---|---|---|---|---|
| Span-ASTE | 0.8579 | 1.1131 | 0.5368 | 0.6597 |
| **Dual-Span** | 0.4443 | 0.5587 | 0.2472 | 0.3169 |

Table 5: Experimental results of time consumption (second) to generate spans on the D2 dataset.

on the D2 dataset with F1 metric and the results are shown in the table 4. On ATE task, Dual-Span is consistently superior to Span-ASTE and GTS, indicating that syntactic and part-of-speech correlation based candidate reduction and representation are effective for aspect term identification. However, on the OTE task, our model is slightly inferior to Span-ASTE on most of the benchmark datasets, which is caused by lower P values. We expect the reason behind lies in the part-of-speech based span generation. In effect, we only consider the spans involving words that are tagged with $JJ$ or $NN$. However, opinion terms can be tagged with $VBN$, which we do not include. We leave the expansion of more valid part-of-speech spans in the future work.

In order to verify that our proposed dual-channel span generation strategy can noticeably reduce the computational cost of span enumeration, we test the time consumption of Dual-span and Span-ASTE on span enumeration under the same runtime environment. From the results shown in Table 5, we can see that the proposed dual-channel span generation strategy cuts time cost in half.

## 5 Conclusion

In this work, we present a Dual-Span model for improving the performance on ASTE task. Based on the observations of syntactic relations and part-of-speech features among spans, we design a dual-channel span generation method to refine the span candidate set so as to mitigate the negative impacts of invalid spans. Moreover, we employ relational graph neural networks to capture the high-order interactions between possible spans from both views: syntactic dependency relation and part-of-speech relation. Our experimental results demonstrate that the proposed method brings meaningful gains to ASTE as well as ATE task, compared to all baselines. We also note that for OTE task, our method is generally inferior to the vanilla span-based method that enumerate all possible spans. The reason may lie in the limited part-of-speech relations, which will be considered in the future work.

## Acknowledgements

This work is supported by National Nature Science Foundation (NO.62276099) and SWPU Innovation Foundation (NO.642).

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

# A  Appendix

## A.1  Dataset statistics

We counted the data of the two versions separately [2] [3]. As shown in the Table 6, in addition to the number of sentences and triples in the dataset, the number of words and multi-word spans is also counted. In addition, we perform part-of-speech statistics on the D1 and D2 data sets. The results are shown in Table 7. The number of parts of speech with $JJ$ or $NN$ accounts for a relatively high proportion of the overall $A\&O$, and the distribution of the rest of the parts of speech is scattered and unrepresentative. As there is a trade-off between prediction accuracy and time consumption, we only consider spans involving words tagged with $JJ$ or $NN$ in Section 4.5.2.

## A.2  Hyperparameter analysis

Figure 4 shows the sensitivity analysis of the hyperparameters $S_{window}$, $L_s$, on the D2 dataset. From the figure, we can observe that the effect is the best when the part-of-speech window $S_{window}$ is 3. In fact, when the part-of-speech window is set to $S_{window} = 3$, it can basically cover all aspect terms and opinion terms whose parts-of-speech are $NN$ and $JJ$. When $S_{window}$ exceeds 3, more noise and complexity may be introduced. When the hyperparameter $L_s$=8, the performance is the best.

## A.3  Impact of SynGAT and PosGAT Layers

To explore the impact of the number of layers of SynGAT and PosGAT in Dual-Span, we evaluate the number of layers of SynGAT and PosGAT on the D2 dataset, where multiple layers indicate that node information can be propagated to higher-order neighbors. As shown in Figure 6, our model achieves the best performance when both SynGAT and PosGAT are two layers. Specifically, on the syntactic dependency tree of a sentence, 2-hops are helpful for the interaction between aspect and opinion terms, while on the part-of-speech graph, 2-hop relations involving $NN$ or $JJ$ are conducive for capturing valid spans. Note that, the performance declines as the Dual-RGAT goes deeper, which may be due to the oversmoothing (Li et al., 2018) of graph neural networks.

---

[2] https://github.com/xuuuluuu/
SemEval-Triplet-data
[3] https://github.com/xuuuluuu/
SemEval-Triplet-data/tree/master/
ASTE-Data-V2-EMNLP2020

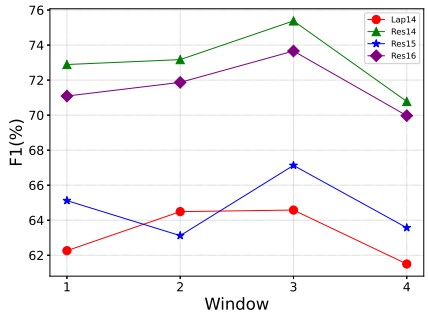

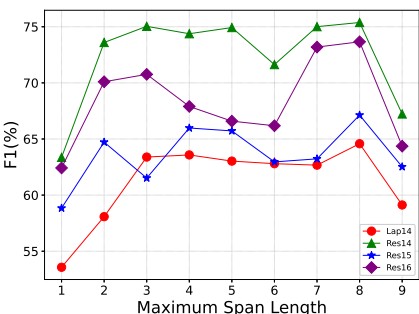

Figure 4: Sensitivity analysis of the hyperparameters $S_{window}$ and $L_s$ on dataset D2.

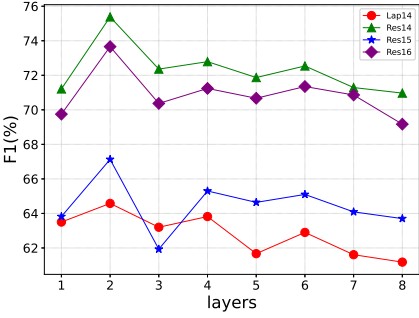

Figure 5: Performance v.s the number of layers in graph networks.

## A.4  Visualization of Linguistic Correlations

To explore how the syntactic dependency and part-of-speech correlation between words contribute to valid span generation, we visualize the attention scores of the syntactic relations and part-of-speech adjacency matrix in two RGAT modules, where rows are queries and columns are keys. As shown in the figure 6, the sampled text "Good creative rolls !" (i.e., the fourth example in the section A.5) contains the triplet: ("rolls", "good creative", "positive"). Since our model employs syntactic dependency relation to learn representations, and exploits the part-of-speech information around indicative words (e.g., the words with "NN" tag) as well, the inter-span and intra-span relations can be success-

| Dataset | | Lap14 | | | | Res14 | | | | Res15 | | | | Res16 | | | |
|---|---|---|---|---|---|---|---|---|---|---|---|---|---|---|---|---|---|
| | | #S | #T | #SW | #MW | #S | #T | #SW | #MW | #S | #T | #SW | #MW | #S | #T | #SW | #MW |
| D1 | Train | 899 | 1452 | 815 | 637 | 1259 | 2356 | 1614 | 742 | 603 | 1038 | 696 | 342 | 863 | 1421 | 931 | 490 |
| | dev | 225 | 383 | 213 | 170 | 315 | 580 | 386 | 194 | 151 | 239 | 165 | 74 | 216 | 348 | 232 | 116 |
| | test | 332 | 547 | 291 | 256 | 493 | 1008 | 674 | 334 | 325 | 493 | 303 | 190 | 328 | 525 | 351 | 174 |
| D2 | Train | 906 | 1460 | 824 | 636 | 1266 | 2338 | 1586 | 752 | 605 | 1013 | 678 | 335 | 857 | 1394 | 918 | 476 |
| | dev | 219 | 346 | 190 | 156 | 310 | 577 | 388 | 189 | 148 | 249 | 165 | 84 | 210 | 339 | 216 | 123 |
| | test | 328 | 543 | 291 | 252 | 492 | 994 | 657 | 337 | 322 | 485 | 297 | 188 | 326 | 514 | 344 | 170 |

Table 6: Statistics for the two versions of the dataset. #S and #T represent the number of sentences and the number of triplets, respectively. #SW indicates that the aspect and opinion terms in the triplets are both single-word spans, while #MW indicates that at least one of the aspect or opinion terms are multi-word spans.

| Dataset | Lap14 | | | Res14 | | | Res15 | | | Res16 | | |
|---|---|---|---|---|---|---|---|---|---|---|---|---|
| | A&O | JJ&NN | Rat | A&O | JJ&NN | Rat | A&O | JJ&NN | Rat | A&O | JJ&NN | Rat |
| D1 | 4447 | 2949 | 0.66 | 7350 | 5944 | 0.81 | 3282 | 2714 | 0.83 | 4262 | 3523 | 0.83 |
| D2 | 4698 | 3113 | 0.66 | 7818 | 6346 | 0.81 | 3494 | 2882 | 0.82 | 4494 | 3707 | 0.82 |

Table 7: Aspect and opinion term part-of-speech statistics on public datasets. Among them, A&O, JJ&NN represent the number of aspect and opinion terms and the number of parts of speech are JJ and NN, respectively, and Rat represents their ratio.

| Review | Ground-truth | Span-ASTE | Dual-Span |
|---|---|---|---|
| The baterry is very longer. | (baterry, longer, P) | (baterry, longer, N) | (baterry, longer, P) |
| And windows 7 works like a charm. | (windows 7, charm, P) | ∅ | (windows 7, charm, P) |
| I wanted it for it 's mobility and man,this little bad boy is very nice. | (mobility, nice, P) | (mobility, wanted, P), (mobility, nice, P) | (mobility, nice, P) |
| Good creative rolls ! | (rolls, good creative, P) | (creative rolls, good, P) | (rolls, good creative, P) |
| The wine list was extensive-though the staff did not seem knowledgeable about wine pairings . | (wine list, extensive, P), (staff, not seem knowledgeable, N) | (wine list, extensive, P) | (wine list, extensive, P), (staff, not seem knowledgeable, N) |
| for 7 years they have put out the most tasty, most delicious food and kept it that way ... | (food, tasty, P), (food, delicious, P) | (food, delicious, P) | (food, tasty, P), (food, delicious, P) |

Table 8: Case study on dataset D2.

| Review | Ground-truth(part of speech) | Dual Span(part of speech) |
|---|---|---|
| The OS is fast and fluid, everything is organized and it 's just beautiful. | fast(JJ), organized(VBN), fluid(NN), beautiful(JJ) | fast(JJ), fluid(NN), beautiful(JJ) |
| This place has ruined me for neighborhood sushi. | ruined(VBN) | ∅ |
| I think the pizza is so overrated and was under cooked. | overrated(JJ), under cooked(IN, VBN) | overrated(JJ) |
| Decor needs to be upgraded but the food is amazing! | upgraded(VBN), amazing(JJ) | amazing(JJ) |

Table 9: A case study of prediction errors in OTE tasks on the D2 dataset.

| Review | A&O(part of speech) | Ground-truth | Dual-Span |
|---|---|---|---|
| The OS is fast and fluid, everything is organized and it 's just beautiful. | os(NNP), fast(JJ), fluid(NN), organized(VBN), beautiful(JJ) | (os, fast, P), (os, organized, P), (os, fluid, P), (os, beautiful, P) | (os, fast, P), (os, fluid, P), (os, beautiful, P) |
| This place has ruined me for neighborhood sushi. | sushi(NN), ruined(VBN) | (sushi, ruined, P) | ∅ |
| I think the pizza is so overrated and was under cooked. | pizza(NN), overrated(JJ), under cooked(IN, VBN) | (pizza, overrated, N), (pizza, under cooked, N) | (pizza, overrated, N) |
| Decor needs to be upgraded but the food is amazing! | 'decor(NN), food(NN), upgraded(VBN), amazing(JJ) | (decor, upgraded, N), (food, amazing, P) | (food, amazing, P) |
| The manager was rude and handled the situation extremely poorly. | manager(NN), rude(VBN) | (manager, rude, N) | (manager, rude, N), (manager, poorly, N), (situation, poorly, N) |

Table 10: A case study of prediction errors on the D2 dataset.

fully captured by graph attention networks. So the aspect term "rolls" pays attention to opinion terms "good" and " creative" (the last row of the left plot in Figure 6), while the two words with "JJ" tag, i.e., "good" and " creative" shows more strong correlation than with "rolls" in part-of-speech graph (corresponding to the right plot of Figure 6), demonstrating that they are more likely to fall into the same category. As a result, our model gives the correct triplet, in contrast to Span-ASTE whose prediction is ("creative rolls", "good", "positive").

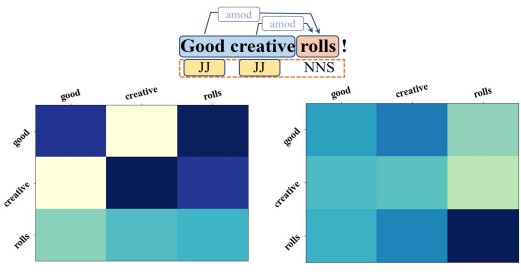

(a) Review: "Good creative rolls!"

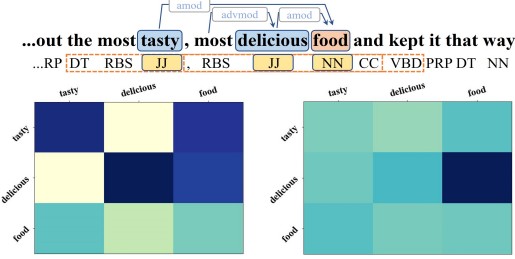

(b) Review: "for 7 years they have put out the most tasty, most delicious food and kept it that way..."

Figure 6: Visualization of adjacency tensors of syntactic (left) and part-of-speech combination features (right).

## A.5 Case Study

We use several examples from the test set of dataset D2 to analyze and validate our method, as shown in the Table 8. For the first example, our method Dual-Span may perform better in predicting sentiment consistency than Span-ASTE. On the 2nd,

5th, and 6th examples, it can be shown that our method makes full use of syntactic and semantic information to improve the accuracy of effective span capture. It can be shown in the fourth example that our method reduces the span boundary error by using part-of-speech structural features.

## A.6 Error analysis

In order to explore the reason behind the slight inferiority of our model on the OTE task, compared to Span-ASTE on most benchmark datasets, we conduct error case study analysis on the datasets of the D2 version. As shown in Table 9, since we do not use the tag "$VBN$" in constructing part-of-speech graph, our method fails to extract the opinion words with the part of speech "$VBN$" in the four examples, e.g. "organized ($VBN$)" and "upgraded($VBN$)". Additionally, to identify the limitations of our work and potential areas for improvement in the future, we perform error sample analysis on the D2 dataset. As shown in Table 10, for opinions whose part of speech is not $JJ$, our method is more likely to give wrong prediction results. Moreover, there are some non-aspect or opinion words whose part-of-speech are $NN$ or $JJ$, which also mislead the model to make wrong span identification.