# OpenReview forum: "Dual-Channel Span for Aspect Sentiment Triplet Extraction"
_EMNLP/2023/Conference — EMNLP 2023 Main_

### Official Review · Reviewer_V7Ee · 2023-07-24

**Soundness:** 3

**Excitement:**

4: Strong: This paper deepens the understanding of some phenomenon or lowers the barriers to an existing research direction.

**Justification For Ethical Concerns:**

None.

**Missing References:**

-For ABSA:
Kim Schouten, Flavius Frasincar: Survey on Aspect-Level Sentiment Analysis. IEEE Trans. Knowl. Data Eng. 28(3): 813-830 (2016)

-For AD:
Maria Mihaela Trusca, Flavius Frasincar: Survey on Aspect Detection for Aspect-Based Sentiment Analysis. Artif. Intell. Rev. 56(5): 3797-3846 (2023)

-For ABSC:
Gianni Brauwers, Flavius Frasincar: A Survey on Aspect-Based Sentiment Classification. ACM Comput. Surv. 55(4): 65:1-65:37 (2023)

**Paper Topic And Main Contributions:**

The paper presents an end-to-end approach for (aspect, opinion, sentiment) triplet extraction based on RGAT and span filtering using two graphs: dependency graph and POS graph. The results show that the proposed approach is superior to existing ones for the triplet extraction task and both graphs complement each other and are useful for the considered task.

The paper is in general well-written, and the proposed solution seems original. I particularly appreciate the extraction of useful spans that increase the efficiency compared to a brute-force approach. The evaluation is thorough and shows the usefulness of the proposed method.

**Questions For The Authors:**

1. Why do you say in the complexity analysis that spans are composed only of two words?
2. What is the difference between a syntax tree and part-of-speech tree?
3. Are the times to construct all spans below 1 second?

Thank you for the rebut, which appropriately answers my questions.

**Reasons To Accept:**

+ well written paper
+ novel solution
+ thorough evaluation

**Reasons To Reject:**

- some references missing
- some explanation missing
- missing link to code

**Reproducibility:**

3: Could reproduce the results with some difficulty. The settings of parameters are underspecified or subjectively determined; the training/evaluation data are not widely available.

**Reviewer Confidence:**

4: Quite sure. I tried to check the important points carefully. It's unlikely, though conceivable, that I missed something that should affect my ratings.

**Typos Grammar Style And Presentation Improvements:**

-page 1: "for" instead of "For" (in title)
-page 2: “some frequent patterns” instead of “some frequent pattern”
-page 3: “GloVe” instead of “Glove”
-page 4: “correlation, incorporating” instead of “correlation. Incorporating”
-page 4: “each word” instead of “each Word”
-page 5: “),” instead of “) ,”
-page 6: “BERT” instead of “bert”

---

> ### Author Rebuttal · Authors · 2023-08-28
>
> Comment 1.Why do you say in the complexity analysis that spans are composed only of two words?
>
> Reply: Thank the reviewer for carefully checking our manuscript. Referring to Section 3.4.1 in the manuscript, Formula 7, the representation of span is composed of the beginning and ending words and the distance between them, so only the first and last words are necessary to be predicted. Therefore, time complexity only involves the computation of the beginning and ending words.
>
> Comment 2.What is the difference between a syntax tree and part-of-speech tree?
>
> Reply: We do not mention “syntax tree” and part-of-speech tree in the text. We use the syntactic dependency tree. In addition, we leverage part-of-speech tags to construct a part-of-speech graph, which is not a “tree”, as there are cycles in this graph.
>
> Comment 3. Are the times to construct all spans below 1 second?
>
> Reply: Yes. Span generation in our method takes time less than 1 second, and less than that of Span-ASTE by half.
>
> Missing References:
>
> Reply: We appreciate the reviewer's suggestions. We have reviewed the recommended papers and included them as references in the “related work” section, and discussed recent trends in the field of sentiment analysis in the revised version.
>
> Typos Grammar Style And Presentation Improvements:
>
> Reply: We have carefully checked the manuscript and corrected all the errors pointed out by the reviewer.

---

### Official Review · Reviewer_khna · 2023-08-05

**Soundness:** 4

**Excitement:**

4: Strong: This paper deepens the understanding of some phenomenon or lowers the barriers to an existing research direction.

**Paper Topic And Main Contributions:**

The main content of the article is the proposal of a dual-channel span generation approach called Dual-Span for aspect-level sentiment triplet extraction. The approach utilizes two relational graph attention networks to learn high-order syntactic dependency between words/spans and linguistic features in constructed part-of-speech relations among words. The article also discusses the lexical characteristics of aspect and opinion terms and how they can be used to extract aspect/opinion spans.

The main contribution of the article is the proposal of the Dual-Span approach, which achieves state-of-the-art performance on the aspect-level sentiment triplet extraction task. The approach is shown to outperform existing methods that lack a complete consideration of the relationship between the three sentiment elements. The article also provides insights into the lexical characteristics of aspect and opinion terms, which can be useful for future research in this area.

**Questions For The Authors:**

Here are my questions:
1.Why did you choose to compare the proposed method with only span-based approaches and not with non-span-based approaches, such as sequence labeling or joint modeling? Do you think the proposed method would perform better or worse than these approaches?

 2.Could you provide a detailed analysis of the errors made by the proposed method? What are the limitations and potential areas for improvement of the approach?

**Reasons To Accept:**

The strengths of this paper include:

1. Novel approach: The proposed Dual-Channel Span method is a novel approach to ASTE task that reduces the search space of span candidates and improves the performance of sentiment triplet extraction. The approach generates aspect and opinion spans separately and then combines them to form aspect-opinion pairs, which is different from existing span-based approaches.

2. State-of-the-art performance: The approach is evaluated on two public datasets and achieves state-of-the-art performance, demonstrating its effectiveness and superiority over existing span-based approaches. This is a significant contribution to the field of fine-grained aspect-based sentiment analysis.

3. Clear presentation: The paper is well-written and clearly presents the proposed method, the experimental setup, and the results. The authors provide detailed analysis and discussion of the results, which makes it easy for readers to understand the contributions and limitations of the approach.


**Reasons To Reject:**

1.  Limited evaluation: The proposed method is evaluated on only two public datasets, which may not be sufficient to demonstrate its effectiveness and generalizability to other domains and languages.  The authors could consider evaluating the approach on more diverse datasets to address this limitation.   2.  Lack of comparison with non-span-based approaches: The authors compare the proposed method with existing span-based approaches, but do not compare it with non-span-based approaches, such as sequence labeling or joint modeling.  This may limit the scope of the evaluation and the potential impact of the approach.   3.  Lack of analysis of errors: The authors do not provide a detailed analysis of the errors made by the proposed method, which could help identify the limitations and potential areas for improvement.  The authors could consider providing such analysis in future work.

**Reproducibility:**

4: Could mostly reproduce the results, but there may be some variation because of sample variance or minor variations in their interpretation of the protocol or method.

**Reviewer Confidence:**

3: Pretty sure, but there's a chance I missed something. Although I have a good feel for this area in general, I did not carefully check the paper's details, e.g., the math, experimental design, or novelty.

---

> ### Author Rebuttal · Authors · 2023-08-28
>
> Comment 1. Limited evaluation: The proposed method is evaluated on only two public datasets, which may not be sufficient to demonstrate its effectiveness and generalizability to other domains and languages. The authors could consider evaluating the approach on more diverse datasets to address this limitation.
>
> Reply: Thank to the reviewer for raising this issue. In the field of aspect-level sentiment analysis, the public datasets that contain parsing tags and triplet labels are quite limited and all included in the benchmarks we use (i.e., Lap14, Res14, Res15, Res16). Due to limited time, we will leave the supplementary experiments in the future work.
>
> Comment 2. Lack of comparison with non-span-based approaches: The authors compare the proposed method with existing span-based approaches, but do not compare it with non-span-based approaches, such as sequence labeling or joint modeling. This may limit the scope of the evaluation and the potential impact of the approach.
>
> Reply: Thank the reviewer for this comment. In our original manuscript, we have the comparisons between our method and non-span-based approaches (i.e., Table 1 and Table 2 in the original manuscript). Specifically, we compare with two classes of non-span-based approaches: sequence markers (i.e., GTS, JET, S3E2, EMC-GCN), and joint modeling methods (i.e., Li-unified-R, Peng-two-stage, IMN+IOG). The results of these methods are reported in Table 1 and Table 2 in our paper.
>
> Comment 3.  Lack of analysis of errors: The authors do not provide a detailed analysis of the errors made by the proposed method, which could help identify the limitations and potential areas for improvement. The authors could consider providing such analysis in future work.
>
> Reply: We are grateful to the reviewer for this helpful comment. We agree that analysis of the errors made by our method is helpful, which we also demonstrate in the conclusion section. Here we present some wrong cases to clarify our statements in the manuscript. As shown in the table below, for opinions whose part of speech is not JJ, our method is more likely to give wrong prediction results. Moreover, there are some non-aspect or opinion words whose part-of-speech are NN or JJ, which also mislead the model to make wrong span identification.
>
>
> | Review | A&O(part of speech) | Ground-truth |  Dual-Span |
> | -------- | -------- | -------- | -------- |
> | The OS is fast and fluid, everything is organized and it 's just beautiful. | os(NNP), fast(JJ), fluid(NN), organized(VBN), beautiful(JJ) | (os, fast, P), (os, organized, P), (os, fluid, P), (os, beautiful, P) | (os, fast, P), (os, fluid, P), (os, beautiful, P) |
> | This place has ruined me for neighborhood sushi.    | sushi(NN), ruined(VBN)     | (sushi, ruined, P)      | { }    |
> | I think the pizza is so overrated and was under cooked.    | pizza(NN), overrated(JJ),  under cooked(IN, VBN)     | (pizza, overrated, N),(pizza, under cooked, N)     | (pizza, overrated, N)     |
> | Decor needs to be upgraded but the food is amazing!    | decor(NN), food(NN), upgraded(VBN), amazing(JJ)     | (decor, upgraded, N), (food, amazing, P)     | (food, amazing, P)    |
> | The manager was rude and handled the situation extremely poorly.     | manager(NN), rude(VBN)     | (manager, rude, N)    | (manager, rude, N), (manager, poorly, N),  (situation, poorly, N)     |

---

### Official Review · Reviewer_vjaB · 2023-08-05

**Typos Grammar Style And Presentation Improvements:** None
**Soundness:** 3

**Excitement:**

2: Mediocre: This paper makes marginal contributions (vs non-contemporaneous work), so I would rather not see it in the conference.

**Missing References:**

None

**Paper Topic And Main Contributions:**

This paper propose a dual-channel span generation method to reduce span candidates number for alleviating noise in sentiment triplet extraction in Aspect Sentiment Triplet Extraction. Specifically, this paper utilize two relational graph attention networks(RGAT) to learn high-order syntactic dependency between words/spans and linguistic features among words. Then, fusing the syntactic and lexical information to enrich span representation for effectively reducing the noisy information of valid span candidates.

**Questions For The Authors:**

The same to the section "Reasons To Reject"

**Reasons To Accept:**

1.Extensive experiments are conducted and the results can evaluate the effectiveness of the proposed model.

2.The code will be released  after acceptance.

**Reasons To Reject:**

1.The motivation of the paper is much unclear. Specifically, what does  the “higer-order interaction” refer to?  And how does the multiple dependency transmit the higer-order interactions between spans?  How does “indirect relation” suggest the relevance of the sentiment polarity between spans? And how to define the “relevance” between spans? Moreover, only one case is given in the paper and how to evaluate the generalization of the observations  by the authors?

2.The content of “Related work”is incomplete. First, there are many repetitions between “Introduction” and “Related work”. Second, a conclusion of the difference and advantages in this work should be represent clearly compared with prior work in Section 2“Related work”.  Third, the“span-level” ASTE task should be detailed introduction in “Related work”since this work focus on alleviating the span-level ASTE noise problem.

3.It’s not clear what does the “Dual-GAT” mean? and what is the difference between "Dual-GAT" and “Dual-RGAT”. The reason for the replacing  “Dual-RGAT” with Transformer is not clear.

4.More experimental analysis should be added to evaluate the following points: 1) “However, on the OTE task, our model is slightly inferior to Span-ASTE on most of the benchmark datasets”in Section 4.5.2 and 2) “Effectiveness of Dual-Span in Span Generation” and “we only consider the spans involving words that are tagged with JJ or NN”.





**Reproducibility:**

2: Would be hard pressed to reproduce the results. The contribution depends on data that are simply not available outside the author's institution or consortium; not enough details are provided.

**Reviewer Confidence:**

4: Quite sure. I tried to check the important points carefully. It's unlikely, though conceivable, that I missed something that should affect my ratings.

---

> ### Author Rebuttal · Authors · 2023-08-28
>
> Comment 1.The motivation of the paper is much unclear. Specifically, what does the “higer-order interaction” refer to? And how does the multiple dependency transmit the higer-order interactions between spans? How does “indirect relation” suggest the relevance of the sentiment polarity between spans? And how to define the “relevance” between spans? Moreover, only one case is given in the paper and how to evaluate the generalization of the observations by the authors?
>
> Reply：Thank the reviewer for the comments. “Higher-order interaction” refers to the interaction (i.e., message passing) between two distant words in the syntactic dependency graph via multiple hops. For instance, the direct dependency between words is first-order interaction while other indirect dependencies belong to higher-order interaction. So it is clear that higher-order interaction between spans/words will be recursively transmitted by the direct neighbors of the words. The relevance is then reflected by the degree of the interaction between words. Indirect interaction generally suggests some relevance weaker than that of direct interaction. Regarding “how to define the relevance”, the relevance between spans means that there exits syntactic dependency between spans. We have elaborated the above points in the revised manuscript.
>
> Comment 2.The content of “Related work”is incomplete. First, there are many repetitions between “Introduction” and “Related work”. Second, a conclusion of the difference and advantages in this work should be represent clearly compared with prior work in Section 2“Related work”. Third, the“span-level” ASTE task should be detailed introduction in “Related work”since this work focus on alleviating the span-level ASTE noise problem.
>
> Reply：We agree that there are some repetitions between introduction and related work, since the introduction summarizes the existing work and their core ideas, which have been repeated in the “related work”. We demonstrate the difference and advantages of this work compared with prior work in the “Introduction”, not in the “related work”. We appreciate the suggestions from the reviewer for improving the writing.
>
> Comment 3.It’s not clear what does the “Dual-GAT” mean? and what is the difference between "Dual-GAT" and “Dual-RGAT”. The reason for the replacing “Dual-RGAT” with Transformer is not clear.
>
> Reply: We feel sorry for ignoring the explanation of “Dual-GAT”. Compared to Dual-RGAT, Dual-GAT performs graph attentional convolution on syntactic dependency graph and part-of-speech graph respectively without differentiating edge types. As to “Transformer replacing Dual-RGAT”, the reason is that one can evaluate the role of syntaxial relations between words by comparing the performances of specific relation-based learning (i.e., Dual-RGAT based on syntaxial graphs) and complete-graph based learning (i.e. Transformer without considering syntactics and POS) .
>
> Comment 4.1 More experimental analysis should be added to evaluate the following points: 1) “However, on the OTE task, our model is slightly inferior to Span-ASTE on most of the benchmark datasets”in Section 4.5.2.
>
> Reply: Thank the reviewer for the valuable comments. We have incorporated the error case study in Section 4.5.2 in the revision. As shown in table below, since we do not use the tag “VBN” in constructing part-of-speech graph, our method fails to extract the opinion words with the part of speech "VBN" in the four examples, e.g. “organized” (VBN) and “upgraded”(VBN). Accordingly, our method is slightly inferior to Span-ASTE on the OTE task, because we only consider the spans involving words that are tagged with “JJ” or “NN”, while ASTE enumerates all possible spans with the maximum coverage.
>
> | Review | Ground-truth(part of speech)| Dual Span(part of speech)|
> | -------- | -------- | -------- |
> | The OS is fast and fluid, everything is organized and it 's just beautiful. | fast(JJ), fluid(NN), organized(VBN), beautiful(JJ) | fast(JJ), fluid(NN), beautiful(JJ) |
> | This place has ruined me for neighborhood sushi.    | ruined(VBN)     | { }    |
> | I think the pizza is so overrated and was under cooked.    | overrated(JJ),  under cooked(IN, VBN)     | overrated(JJ)    |
> | Decor needs to be upgraded but the food is amazing!    | upgraded(VBN), amazing(JJ)   | amazing(JJ)     |
>
> Comment 4.2 More experimental analysis should be added to evaluate the following points: 2) “Effectiveness of Dual-Span in Span Generation” and we only consider the spans involving words that are tagged with "JJ" or "NN".
>
> Reply:As for the choice of “JJ” and “NN” in our experiments, in fact, there is a tradeoff between prediction accuracy and time consumption, that is, more tags means more time. So we statistically analyze the frequencies of different tags in the datasets. As shown in table below, among them, A\&O, JJ&NN represent the number of aspects and opinion items, respectively, the number of part of speech is "JJ" or "NN", and Rat represents their ratio. The parts of speech of “JJ” and “NN” account for the majority of aspect and opinion, and the rest of the parts of speech are not representative. This is why we adopt “JJ” and “NN”.
>
> | Dataset  | Lap14 |       |      | Res14 |       |      | Res15 |       |      | Res16 |       |      |
> | -------- | ----- | ----- | ---- | ----- | ----- | ---- | ----- | ----- | ---- | ----- | ----- | ---- |
> |          | A&O   | JJ&NN | Rat  | A&O   | JJ&NN | Rat  | A&O   | JJ&NN | Rat  | A&O   | JJ&NN | Rat  |
> | Dataset1 | 4447  | 2949  | 0.66 | 7350  | 5944  | 0.81 | 3282  | 2714  | 0.83 | 4262  | 3523  | 0.83 |
> | Dataset2 | 4698  | 3113  | 0.66 | 7818  | 6346  | 0.81 | 3494  | 2882  | 0.82 | 4494  | 3707  | 0.82 |
>
> Statement: The code has been uploaded to GitHub

---

### Meta-Review · Area_Chair_KXG5 · 2023-09-20

**Recommendation:** 4

**Metareview:**

This paper proposes a dual-channel span generation approach to address the Aspect Sentiment Triplet Extraction (ASTE) task, where two relational graph attention networks are used, namely, syntactic dependency graph and part-of-speech graph. The paper is well written. The solution is reasonable and can help reduce the span candidates. Experiments show the effectiveness of the proposed approach, and can achieve state-of-the-art performance especially on aspect term extraction (ATE) task, while opinion term extraction (OTE) can be improved. The paper uses a lot of acronyms without introduction, e.g., Dual-GAT as pointed out by reviewers. Same for SGAT and PGAT. Please fix all of them by explaining what they mean in the paper. Please add error analysis and improve the related work section. Also please include the responses to reviewers to the paper to make the unclear part clear.

---

### Decision · Program_Chairs · 2023-10-07

**Decision:**

Accept-Main

**Comment:**

This paper proposes a dual-channel span generation approach to address the Aspect Sentiment Triplet Extraction (ASTE) task, where two relational graph attention networks are used, namely, syntactic dependency graph and part-of-speech graph. The paper is well written. The solution is reasonable and can help reduce the span candidates. Experiments show the effectiveness of the proposed approach, and can achieve state-of-the-art performance especially on aspect term extraction (ATE) task, while opinion term extraction (OTE) can be improved. The paper uses a lot of acronyms without introduction, e.g., Dual-GAT as pointed out by reviewers. Same for SGAT and PGAT. Please fix all of them by explaining what they mean in the paper. Please add error analysis and improve the related work section. Also please include the responses to reviewers to the paper to make the unclear part clear.